# Defect Identification Method for Transformer End Pad Falling Based on Acoustic Stability Feature Analysis

**DOI:** 10.3390/s23063258

**Published:** 2023-03-20

**Authors:** Shuai Han, Bowen Wang, Sizhuo Liao, Fei Gao, Mo Chen

**Affiliations:** 1China Electric Power Research Institute, Beijing 100192, China; 2State Grid Zhejiang Electric Power Research Institute, Hangzhou 310014, China

**Keywords:** the end pad falling defect, Mel time–frequency spectrum, time-series spectral entropy, stability, defect identification

## Abstract

A transformer’s acoustic signal contains rich information. The acoustic signal can be divided into a transient acoustic signal and a steady-state acoustic signal under different operating conditions. In this paper, the vibration mechanism is analyzed, and the acoustic feature is mined based on the transformer end pad falling defect to realize defect identification. Firstly, a quality–spring–damping model is established to analyze the vibration modes and development patterns of the defect. Secondly, short-time Fourier transform is applied to the voiceprint signals, and the time–frequency spectrum is compressed and perceived using Mel filter banks. Thirdly, the time-series spectrum entropy feature extraction algorithm is introduced into the stability calculation, and the algorithm is verified by comparing it with simulated experimental samples. Finally, stability calculations are performed on the voiceprint signal data collected from 162 transformers operating in the field, and the stability distribution is statistically analyzed. The time-series spectrum entropy stability warning threshold is given, and the application value of the threshold is demonstrated by comparing it with actual fault cases.

## 1. Introduction

A power transformer is one of the most important pieces of equipment in a power grid. As the current flows through the winding coil during its operation, an electromagnetic force is generated in the leakage magnetic field, resulting in winding vibration. Several factors, such as a reduction in the insulation layer, the short-circuit impact current and pad falling, can lead to winding looseness, causing equipment vibration and changes in the corresponding acoustic time–frequency characteristics, known as the acoustic signature. Analyzing the acoustic feature information of the equipment under different conditions can serve as a crucial basis for determining the transformer’s condition [1,2].

The occurrence and transmission mechanisms of the internal vibration of a transformer are complex [3]. In order to extract vibration characteristic information effectively, researchers usually start from the mathematical model of transformer vibration to mine and analyze winding vibration acoustic signals. The spring–mass pad equivalent model [4] has a wide range of applications and is gradually optimized [5]. A mathematical model of the axial vibration of windings was established based on this equivalent model in Reference [6], and the mechanical characteristics of the spacers were analyzed. It was found that the maximum axial displacement of the end winding under different winding pre-tightening forces increases with an increase in the peak short-circuit current, while the force acting on the end plate of the winding decreases. Reference [7] treated the end pad as an elastic element and constructed a transformer winding coil model. It was found that, when the end pad falling fault occurs, the vibration amplitude of the transformer continues to develop until the end of the winding collides with the iron yoke in a non-elastic manner. The average duration of this process is about 0.680 s.

Based on an analysis of the physical theoretical model, it is necessary to use appropriate mathematical methods to extract the characteristics of the defect [8,9]. Reference [10] constructed models for the radial vibration and oil propagation of windings. The radial and axial vibration characteristics of the winding and pressure plate were experimentally studied using a laser vibrometer and vibration acceleration sensors. The conditions for the generation of multiple-frequency vibration of the winding were derived, providing a reference basis for the diagnosis of faults in transformer windings [11]. Ma Hongzhong et al. investigated the effects of different tightening forces on the vibration signals of transformers using a finite element analysis, focusing on the characteristics of the 100 Hz fundamental frequency vibration signal [12]. They concluded that the inherent frequency of the transformer decreases as the pre-tightening force decreases, which can be used to assess the degree of winding looseness [13]. Hong K proposed a winding condition evaluation model using vibration signals, which separates winding vibration signals from mixed signals using a fundamental frequency vibration analysis [14]. In addition, they used a gated recurrent unit (GRU) neural network to explore the relationship between the current sequence and the vibration sequence for operating transformers, suggesting that the electromagnetic force induced by the load current affects the vibration response of the winding structure, especially when the tightening force is significantly loose [15]. Wang et al. obtained the vibration characteristics of winding under normal and loose conditions from the perspective of pad nonlinearity, and they found a correlation between the degree of winding looseness and the 100 Hz octave component in the vibration signal [16].

The main reason for the exacerbation of winding looseness caused by cumulative stress, such as short-circuit shock in transformers, is the falling of the end pad, which shows quasi-steady-state vibration characteristics and has significant changes in frequency spectrum distribution and amplitude with the time series [17]. Existing research on the mechanical characteristics and spectrum of windings is mostly based on steady-state vibration signals. However, vibration sensors in transformers are often difficult to install in substations, and the acoustic signals transmitted through the oil tank from the internal vibrations of transformers are easier to monitor and can contain more comprehensive information than vibration signals [18,19].

To address the aforementioned problem, a new method is designed in this paper to discriminate the stability level of the acoustic signals of winding vibrations. Firstly, based on the “mass–spring–damping” model, the quasi-steady-state development process of the vibration mode of winding under the condition of end pad falling is analyzed, and a time-series spectral entropy algorithm for signal stability is constructed accordingly. The chaos degree of the acoustic fingerprint time–frequency spectrum during the dynamic change in the vibration signal is described, and the stability of the acoustic fingerprints of a dataset of 162 transformers from 24 substations of the State Grid is calculated. The stability threshold ranges of the two operating states are determined, and the effective identification of end pad falling defects of transformer windings is achieved.

## 2. The Vibration Characteristics Analysis of Winding End Pad Falling

### 2.1. The Analysis of Winding Vibration

In the case of the deformation, displacement or collapse of transformer windings, the compression force between wire cakes is insufficient, thus aggravating the unbalance of winding ampere-turns, and the magnetic leakage generated increases the axial force, which aggravates the winding vibration [20].

According to the structural characteristics of winding, the single-layer coil can be equivalent to a concentrated mass pad, and the insulation gasket can be equivalent to an elastic element. The forced vibration of winding can be equivalent to a dynamic equivalent model of mass–spring–damping, which can better represent the natural vibration characteristics of winding. This is shown in Figure 1a. According to the D. Alembert principle, the dynamic equation of coil elements can be expressed as follows:(1)$$m{\ddot{x}}_{n}+c{\dot{x}}_{n}+kx_{n}=F_{y}+mg$$
where *m* is the mass of the coil element, *c* is the damping coefficient, k is the elastic coefficient of the spring, $x_{n}$ is the displacement of the nth unit coil relative to its original position, and $F_{y}$ is the axial electric force.

In order to further explore the winding vibration response under the condition that the end pad falls off, the top coil is simplified as a single degree of freedom system [21]. In addition, the spring is regarded as a linear system, which balances with the coil mass under working conditions. For an easy calculation, the electrodynamic force is simplified to the form of multiplying the external excitation $F_{y0}$ and the cosine function of twice the grid frequency. The dynamic equation of the top coil is expressed as follows:(2)$$m\ddot{x}\left(t\right)+c_{m}\dot{x}\left(t\right)+k_{m}x\left(t\right)=F_{y0}\cos(2\omega t+2\varphi )$$

Among them, $c_{m}$ and $k_{m}$ are the damping and elastic coefficients between the pressing plate and the iron yoke, respectively. Using the general solution characteristic equation of its homogeneous equation, the characteristic root of the winding damped vibration can be obtained. Without considering the effect of the electrodynamic force, the winding presents an under-damped vibration. The vibration amplitude is written as follows:(3)$$x\left(t\right)=e^{-\zeta \omega_{0}t}\left(c_{1}\cos\omega_{d}t+c_{2}\sin\omega_{d}t\right)$$

Among them, $c_{1}$ and $c_{2}$ are determined by the initial conditions of the winding. The winding has a damped natural frequency, which is shown as $\;\omega_{d}=\omega_{0}\sqrt{1-\zeta^{2}}$. The natural frequency of the winding under undamped vibration $\omega_{0}$ and the relative damping coefficient ζ can respectively be expressed as follows [22]:(4)$$\left\{\begin{array}{c} \omega_{0}=\sqrt{\frac{k_{m}}{m}}\\ \zeta =\frac{c_{m}}{2\sqrt{k_{m}m}} \end{array}\right.$$

The general solution describes the transient process of the system. Equation (7) shows that the winding is subject to damped vibration with the amplitude attenuated gradually. Considering the special solution of the non-homogeneous equation, the variable s is introduced:(5)$$s=\frac{\omega_{F}}{\omega_{0}}$$

Among them, $\omega_{F}$ refers to the frequency when the winding is externally excited. Since the electric force is twice the grid frequency, $\omega_{F}$ has a fixed value of 100 Hz.

According to the mathematical transformation, the amplitude amplification factor *β*(*s*) and the phase difference *θ*(*s*) can be expressed as follows:(6)$$\left\{\begin{array}{l} \beta \left(s\right)=\frac{1}{\sqrt{{\left(1-s^{2}\right)}^{2}+{\left(2\zeta s\right)}^{2}}}\\ \theta \left(s\right)=tg^{-1}\frac{2\zeta s}{1-s^{2}} \end{array}\right.$$

Finally, the amplitude expression of the winding forced vibration when the top coil is under the action of an electrodynamic force is obtained as follows:(7)$$x\left(t\right)=\frac{F_{y0}}{k_{m}}\beta \left(s\right)\cos(2\omega t+2\varphi -\theta )$$

According to the response analysis of the winding forced vibration under electrodynamic excitation, it can be seen that the vibration amplitude of the top winding is periodic, and the vibration frequency is twice the system frequency. Under the condition of constant external excitation, the amplitude *x*(*t*) is affected by the amplitude amplification factor *β*(*s*).

### 2.2. The Vibration Analysis of End Pad Falling

This section analyzes the vibration of the end pad in the state of falling off based on the obtained mathematical expression and the influence factors of the winding vibration amplitude under electrodynamic excitation.

When the transformer operates normally, the insulation pressing plate is placed on the top coil. In order to increase the preload, the insulation pad is added between the pressing plate and the upper iron yoke. When the pad falls off, a small gap is formed between the pressing plate and the upper iron yoke. Considering that the insulation pressing plate has a wooden structure and a low density, the quality of the pressing plate is not considered in the mathematical analysis. The dynamic equivalent model of the wire cake under this fault condition is shown in Figure 1b.

The insulating pad between the wire cakes is a nonlinear material, and its stiffness varies with the change in the preload. In the literature [23,24,25], on the basis of a large number of experiments, it is believed that the relationship between the stress on the insulating pad and the elastic coefficient is expressed as follows:(8)$$k_{m}=\frac{S_{w}}{h}\frac{d\sigma }{d\varepsilon },\sigma =a\varepsilon +b\varepsilon^{3}$$
where *σ* and ε represent the stress and strain of the insulation pad, respectively. *ɑ* = 1.05 × 10^3^ kg/cm^2^, *b* = 1.75 × 10^4^ kg/cm^2^. $S_{w}$ is the contact area between the insulating pad and the wire cake, and h is the height of the insulating pad.

Due to the end pad falling off of the transformer, the contact area $S_{w}$ between the pad and the wire cake is zero, which results in a reduction in the pad elastic coefficient $k_{m}$. The natural vibration frequency in Equation (4) decreases. When the grid frequency is constant, the variable s in Equation (5) increases. According to the response curve in Figure 2, it can be judged that $\beta \left(s\right)$ increases continuously; the vibration of the top coil in Equation (7) shows an over-damped vibration mode, and the amplitude increases. The winding pressure plate is made of a nonlinear material and has an inelastic collision with the iron yoke. So, the vibration amplitude of the winding will be reduced, the energy will be accumulated again, and the cycle will be repeated.

In order to describe the mathematical model intuitively, this paper calculates the natural frequency range and the amplitude amplification factor of the winding vibration by using the parameters of the two-dimensional transformer model in Reference [24]. Assuming that the gap between the on-site pressure plate and the iron yoke is 0.01 m, the amplitude variation rule of the top winding with time is shown in Figure 3. According to the amplitude change, the winding state will alternate in two cases: (1) no collision with the iron yoke and (2) collision with the iron yoke. When there is no collision with the iron yoke, the falling of the end pad causes the wire cake to be over-damped, and the vibration amplitude increases steadily in a short time. When colliding with the iron yoke, the winding will generate a broadband mechanical wave, which forms a broadband acoustic signal. According to the above dynamic change, it can provide a theoretical basis for the acoustic features described below.

## 3. The Time-Series Spectral Entropy Stability Algorithm

According to the analysis of the vibration characteristics, when the pad at the end of the winding falls off, the vibration will appear in a quasi-steady state, with the amplitude increasing steadily and the collision alternating. Therefore, this paper constructs an algorithm to calculate the stability of the acoustic signal using time-series spectrum entropy, which can describe the dynamic characteristics of the vibration spectrum quantitatively, and then the defect diagnosis can be realized.

### 3.1. Calculation Process

First of all, spectrum transformation and feature extraction are carried out for the acoustic signal collected at the site of the end pad falling off fault, and then the time-domain signal is converted into the acoustic time–frequency spectrum.

Secondly, a Mel filter is constructed to convert the acoustic time–frequency spectrum into the Mel time–frequency spectrum to realize the compression perception of the acoustic signal.

Then, by extracting each frame vector of the Mel time–frequency spectrum matrix, obtaining the spectrum difference for the adjacent frame vector and calculating the time-series spectral entropy of the spectrum difference vector, a new time-series spectrum entropy transverse vector is arranged by time.

Finally, the root-mean-square value of the time-series spectrum entropy is calculated to obtain the stability of the entire acoustic time–frequency spectrum. The calculation method is shown in Figure 4. The darker color in the time–frequency spectrum represents stronger vibration energy in the corresponding time-frequency region.

### 3.2. Acoustic Signal Processing

The transformer acoustic signal under the condition of pad falling is intercepted for four seconds. Taking into account the stable characteristics of the acoustic signal in a short time interval, the signal is divided into frames, and windows are added; then, the short-time Fourier transform (STFT) of each frame signal is calculated [25]. The sampling frequency of the signal $f_{s}$ is 48,000 Hz; the frame length and the frame shift are set to be 0.1 times and 0.02 times of the sampling frequency, respectively; and the spectral resolution $f_{0}$ is 10 Hz. In order to reduce signal spectrum leakage and signal distortion effectively, the Blackman–Harris window function is used to process the frame length signal. The Blackman–Harris window formula is
(9)$$W_{N}\left(n\right)=\left\{\begin{array}{c} a_{0}-a_{1}\cos(\frac{2\pi n}{N-1})+a_{2}\cos(\frac{4\pi n}{N-1})\\ -a_{3}\cos(\frac{6\pi n}{N-1}),\;0\leq n\leq N-1\\ 0,\;otherwise \end{array}\right.$$
where $a_{0}$ is 0.35875, $a_{1}$ is 0.48829, $a_{2}$ is 0.1428, and $a_{3}$ is 0.01168, which are the intrinsic parameters of the window, and N is the signal length.

The time-domain signal is converted into an acoustic time–frequency spectrum signal, and the acoustic time–frequency spectrum image is stacked according to the time dimension. The image contains three kinds of signal information: time, frequency and signal strength. From this, it can be seen that the acoustic time–frequency spectrum image is a special data representation that combines the time-domain, frequency-domain and image characteristics.

Taking the real fault of the end pad falling off a 35 kV transformer as an example, the situation of a transformer without a cover is shown in Figure 5, with the red circle indicating the appearance after falling of the end pad. After preprocessing the time-domain signal and short-time Fourier transform, the acoustic time–frequency spectrum of the fault case signal is shown in Figure 6a. Additionally, fast Fourier transform is performed on the time-domain signal to obtain 50 Hz and its frequency doubling, the spectrum distribution of which is shown in Figure 6b.

Figure 6 shows that the spectrum distribution range under the condition of pad falling is wide, and more than 94% of energy is concentrated in the range of 0 Hz~2000 Hz, with the main frequency component being 1300 Hz. Based on the above frequency distribution range, the calculation range of the acoustic signals can be set to 0~2000 Hz. In addition, the acoustic features show the distribution characteristics of steady state and collision transient alternating, which is consistent with the conclusion of the theoretical analysis in Section 2.2.

### 3.3. Acoustic Compression Perception

Considering that the human ear’s perception of the scale within the audible range is nonlinear, in order to facilitate the on-site inspection personnel’s manual auscultation, the signal spectrum is compressed to facilitate acoustic online monitoring [26]. In this paper, a Mel filter group is introduced to reduce the dimension of the acoustic time–frequency spectrum signal and to reduce the weight of the interference frequency band. The corresponding relationship between the actual frequency and Mel’s perceived frequency is described as follows [27]:(10)$$Mel\left(f_{mel}\right)=2595\mathrm{lg}(1+f/700)$$
(11)$$Mel^{-1}\left(f\right)=700\times \left(10^{\left(1+k/2595\right)}-1\right)$$
where f is the frequency of normal scale, which is in the range of 0~2000. $f_{mel}$ is Mel’s perceived frequency. Their units are both Hz.

Traditional Mel filter banks often use the triangle filtering method. To move closer to the frequency characteristics of the transformer vibration signals, this paper selects the Blackman–Harris window to design filter banks. A total of 24 band-pass filters are set in the spectrum range of 2000 Hz. Each Mel filter has the Blackman–Harris window filtering characteristics, and their central frequency is $f\left(m\right)$. In order to ensure that each band-pass filter is of equal width in the Mel spectrum range, the transfer function of each band-pass filter in combination Equation (9) is set as follows:(12)$$H_{m}\left(k\right)=\left\{\begin{array}{c} 0\\ W_{p}\left(k-f\left(m-1\right)\right)\\ W_{q}(k+f\left(m+1\right)\\ -2f\left(m\right))\\ 0 \end{array}\begin{array}{c} ,f_{mel}<f\left(m-1\right)\\ ,f\left(m-1\right)\leq f_{mel}\leq f\left(m\right)\\ ,f\left(m\right)<f_{mel}\leq f\left(m+1\right)\\ ,f_{mel}>f\left(m+1\right) \end{array}\right.$$

Among them, *p* is 2 × {*f*(*m*) − *f*(*m* − 1)}; *q* is 2 × {*f*(*m* + 1) − *f*(*m*)}; and m represents each filter, which is in the range of 0~24. $f\left(m\right)$ represents the center frequency of the filter bank. Its expression is
(13)$$f\left(m\right)=\frac{1}{f_{0}}Mel^{-1}(Mel(f_{min})+m\frac{Mel\left(f_{\max}\right)-Mel\left(f_{\min}\right)}{M+1})$$
where $f_{max}$ and $f_{min}$ represent the maximum and minimum values of the filter range, respectively. In this paper, $f_{max}$ is 2000 Hz, and $f_{min}$ is 0 Hz.

The transfer function in Equation (12) is normalized to obtain the relative amplitude of $H_{m}$, which is shown in Figure 7.

The transfer function matrix size of the Mel filter bank designed in Figure 7 within the frequency range of 2000 Hz is [24 × 2001], and the Mel filter transfer function matrix and the acoustic signal time–frequency spectrum matrix are multiplied. We can finally obtain the time–frequency spectrum matrix under the Mel scale. Figure 8 describes the specific process of converting the acoustic time–frequency spectrum matrix when the pad falls to the Mel time–frequency spectrum matrix.

The acoustic time–frequency spectrum matrix is converted into the Mel time–frequency spectrum matrix, which eliminates the influence of the noise signal while retaining the original signal characteristics. The matrix size is changed from the original [2001 × 196] and compressed into [24 × 196], which is reduced by an order of magnitude. This can reduce the computational complexity of subsequent stability calculations and allow for data samples to be processed more efficiently.

### 3.4. Stability Calculation

This section introduces the calculation expression of the time-series spectral entropy stability.

#### 3.4.1. The Spectrum Difference in Mel Time–Frequency Spectrum Adjacent Frame Vector

The spectrum difference in adjacent frame vectors under the Mel scale is calculated, and it is arranged into a new spectrum difference vector over time:(14)$$X_{i}=x_{i+1}-x_{i},\left(1\leq i\leq T-1\right)$$
where *T* represents the number of time–frequency spectrum frame vectors, which is 196 in this example.

#### 3.4.2. Time-Series Spectral Entropy Algorithm

The Mel time–frequency spectrum entropy feature vector is calculated [28]:(15)$$H_{i}=\frac{1}{M}\mathrm{sgn}(X_{i})\overset{M-1}{\underset{i=1}{\sum }}\log_{2}\left|X_{i}\right|$$
where *M* represents the number of Mel time–frequency spectral lines. In this example, *M* is taken as 24. The sgn function formula is as follows:(16)$$sgn\left(\Delta x\right)=\left\{\begin{array}{c} 1\;\;\;\;\;\;\;\;\;\;\;\;\Delta x>0\;\\ 0\;\;\;\;\;\;\;\;\;\;\;\;\Delta x=0\\ -1\;\;\;\;\;\;\;\;\Delta x<0 \end{array}\right.$$where $X_{i}$ is the ith frame vector of the signal.

Then, the feature vector is normalized:(17)$$Z_{i}=\frac{1}{T-1}\overset{T-1}{\underset{i=1}{\sum }}H_{i}$$

#### 3.4.3. Calculate the Stability

The mean square root of spectrum entropy sequence *Z* is calculated to obtain the stability *V*:(18)$$V=\mathrm{rms}\left(Z\right)=\sqrt{\frac{\sum_{i=1}^{T-1}Z_{i}{}^{2}}{T-1}}$$

The algorithm characterizes the change in the spectral energy concentration of the acoustic signal. When the signal is distributed instantaneously, the larger the difference between the eigenvectors formed by the spectral entropy of adjacent spectral sequences, the weaker its autocorrelation and the smaller the calculated value of *V*. On the contrary, the closer the *V* value is to 1, the stronger the signal correlation and the higher the stability. Therefore, the signal stability can be judged according to the value of *V,* and the fault of pad falling can be diagnosed.

## 4. The End Pad Falling and the Acoustic Sample of the Control Group

In order to obtain sound samples of the end pad falling to verify the validity of the algorithm, this paper conducts an acoustic acquisition experiment of an oil-immersed transformer with the end pad falling off. In addition, the winding is usually in the loose state before the end pad of the transformer falls off, so this paper also sets the corresponding loose state of the winding test samples in addition to the normal state test samples.

The technical parameters of the transformer used for the simulation test are shown in Table 1. The transformer is pressurized to the rated value via the voltage regulator connected to the console so that the transformer can operate without load. This experiment only simulates the looseness fault of phase C of the winding.

To obtain sound samples of the transformer when there is no winding looseness or pad falling off as a control, 30~110% of the rated voltage is applied to the transformer in the experiment, and the applied voltage interval of each group of experiments is 5%. Thus, 17 sets of transformer sample data were obtained.

### 4.1. Simulation Experiment of Winding Looseness Fault

The axial direction of the transformer wire cake is fixed by pressing the upper and lower clamps. The upper and lower clamps are connected and fixed by screws, which limit the axial vibration intensity of the transformer winding from to being too large, and the tightening degree of the pressing nails directly affects the clamping force of the winding. Therefore, this experiment mainly controls the winding clamping force by adjusting the pressing nails. There are eight compression pins on both sides of the iron yoke in the test transformer, which are evenly arranged on both sides of the iron core. Only the looseness of the single-phase winding of the transformer is considered when setting the fault.

In order to fully simulate various loosening conditions and avoid the homogenization of the test samples, simulation experiments with different degrees of loosening are carried out on different pressure nails. Phase C is selected as the loosening object for the test. The schematic diagram of loose arrangement of transformer windings and the corresponding locations and numbers of the pressing nails are shown in Figure 9. The winding loosening test scheme is shown in Table 2, and the transformer is in the rated operation state. The process of winding from normal compaction to complete loosening is divided into 13 stages. The transformation process of the transformer vibration state and the vibration mode is recorded and analyzed. After the test is completed, four compression pins are tightened to make the winding return to the compaction state.

### 4.2. Simulation Experiment of Winding End Pad Falling Fault

In this simulation test, the winding end pad is taken out, and a gap of about 1 cm is artificially created. The method of the winding looseness test is followed, and a simulation of the winding looseness fault when the pad falls off is also conducted, which is close to the state of a real transformer when the winding end pad falls off.

To avoid the homogenization of the test samples, the falling simulation experiment is carried out on different positions and different numbers of pads. The corresponding relationship between the number of end pads falling off and the loosening range is shown in Figure 10. The site layout of pad falling is shown in Figure 11. The defect layout object is still in phase C. The test method shown in Table 3 is the same as that in Section 3.2, and the cumulative loosening test method is adopted to collect the acoustic data under each loosening degree.

## 5. Verification of Time-Series Spectral Entropy Stability Algorithm

In this section, stability V-value calculations are performed on acoustic signature datasets of rated states under experimental conditions, two different fault categories and one real case of the end pad falling fault. To demonstrate the effectiveness of the algorithm, traditional distance measures are used for comparisons and verification. In addition, a set of real samples from transformers in operation is used to provide a normal reference range for time-series spectral entropy stability.

### 5.1. Dataset Stability Calculation

Stability calculations are performed using simulated acoustic signature datasets of transformers with faults under three different conditions and one real case of the end pad falling fault. In addition to time-series spectral entropy, Euclidean distance and cosine distance are used for comparisons. Comparisons of the acoustic spectrograms and the Mel spectrograms under different conditions are shown in Figure 12.

### 5.2. A Comparison of Algorithms

In order to compare the effectiveness of the three stability algorithms in the identification of the pad falling fault, this paper compares the stability calculation results horizontally, and the results are shown in Figure 13.

(1) Euclidean distance stability: the stability distribution range of the rated compression force dataset is 1–99%, and the quantile value is 2.15 × 10^−2^–4.48 × 10^−2^. In comparison, the overall trend of the stability distribution of the winding looseness fault is not obvious, and the 1–99% of quantile value is 2.13 × 10^−2^–5.45 × 10^−2^; the stability of the end pad falling samples is 1–99%, and the quantile value is 3.61 × 10^−2^–6.05 × 10^−2^; and the stability of the actual pad falling fault case is 5.25 × 10^−2^. Therefore, the numerical distribution interval of the stability calculated by using this algorithm is relatively large, and it is unable to distinguish the end pad falling fault.

(2) Cosine distance stability: the difference between the rated compression force dataset and the winding looseness dataset under the algorithm is not significant. The stability distribution ranges are 4.43–8.82 and 4.59–12.27 in the range of 1–99%, respectively. The stability of the end pad falling samples is 10.18–16.22 in the range of 1–99%, and the stability of the actual pad falling fault case is 16.7773. Similar to the Euclidean distance stability, the cosine distance stability cannot distinguish the end pad falling fault.

(3) Time-series spectrum entropy stability: the algorithm calculates the stability from the distribution uniformity of the spectrum signal energy. The stability value of the end pad falling samples is 0.41–1.11, which is lower than the 1% quantile line of the other two types of datasets. The stability calculation results of the rated compression force dataset and the winding looseness dataset are 2.38–2.89 and 1.22–2.53, respectively. It can be seen that this method is effective in distinguishing the end pad falling fault.

To sum up, compared with the Euclidean distance and the cosine distance, the time-series spectrum entropy stability algorithm can ensure that the value of V of the end pad falling fault is outside of the 1–99% quantile line of the two datasets, achieving the identification of this fault.

### 5.3. Samples Data Distribution of 500 kV Transformer in Operation

The time-series spectrum entropy stability algorithm can effectively distinguish winding looseness and pad falling. However, the distribution range of the stability values of large transformers in operation is different from that of the small transformers used for fault simulation in this paper. Therefore, this section further adds a group of acoustic data of 500 kV transformers in operation of the power grid under normal operating conditions to delimit the normal range of stability for the site sample set. This dataset comprises 324 groups of acoustic data of 162 transformers in 24 substations of 500 kV in China in two periods, and the stability statistics of the 324 groups of acoustic data are determined using statistical methods.

As is shown in Figure 14, the time-series spectral entropy of this group of data presents a log-normal distribution with a mean value of 2.293 and a variance of 9.457 × 10^−2^. The sample distribution range is mainly between 1.5321 and 3.1130. So, 1.5321 can be determined to be the warning threshold. The value of the real case of pad falling in this paper is also less than the threshold value, which shows that the threshold value has a certain application value.

## 6. Conclusions

Starting with the transformer pad falling fault, this paper analyzes the vibration process of the fault from the perspectives of mechanisms and acoustics. In view of the high degree of confusion in the acoustic time–frequency spectrum signal of the fault, a stability calculation formula is introduced, which provides a reference value for the determination of the steady-state operation and the non-steady operation of transformers. The main conclusions of this paper are as follows:

(1) According to the electromagnetic vibration analysis theoretical model, when the distance between the winding and the yoke is 10 mm, the winding and the yoke collide every 0.83 s, causing the acoustic signals of the winding to alternate between stable signals and collision signals. This is consistent with the spectral pattern of the acoustic signal sample from the real fault case, which verifies the rationality of the theoretical model proposed in this paper.

(2) The acoustic signal in the matrix form is compressed approximately 83 times using Mel filter banks for data compression perception while retaining the original signal characteristics, which provides support for subsequent feature extraction and stability calculation.

(3) Using the time-scale spectral entropy algorithm for the stability calculation, it is possible to distinguish the samples of pad falling faults in the simulated experimental dataset with 100% accuracy. Furthermore, by conducting voiceprint testing on 162 500 kV transformers, the warning threshold for time-series spectral entropy stability is determined to be 1.5321, which can provide a reference for the diagnosis of pad falling faults in real transformers.

## Figures and Tables

**Figure 1 sensors-23-03258-f001:**
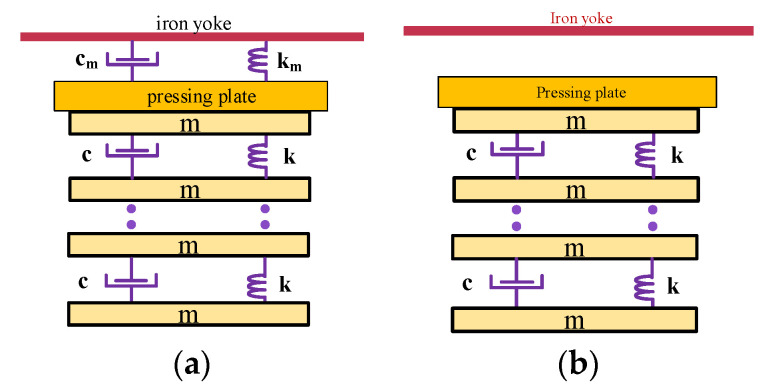
Mass–spring–damping equivalent model. (**a**) Normal model. (**b**) Pad falling model.

**Figure 2 sensors-23-03258-f002:**
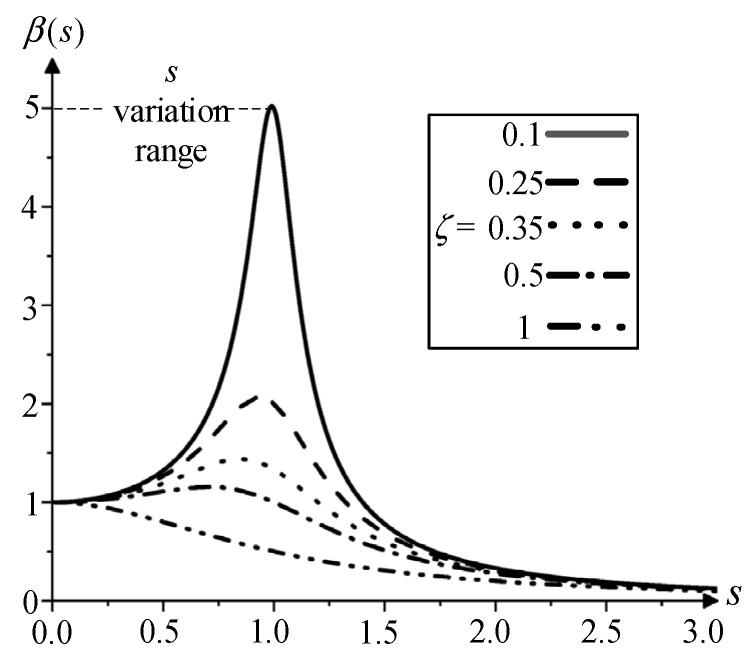
Steady-state response curve of amplitude amplification factor.

**Figure 3 sensors-23-03258-f003:**
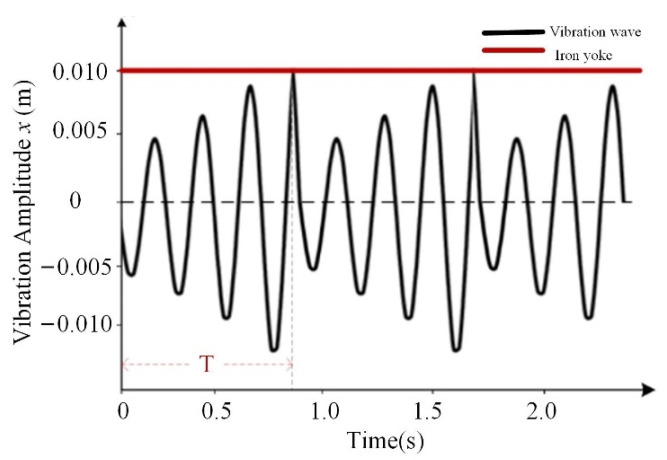
Law of winding vibration amplitude.

**Figure 4 sensors-23-03258-f004:**
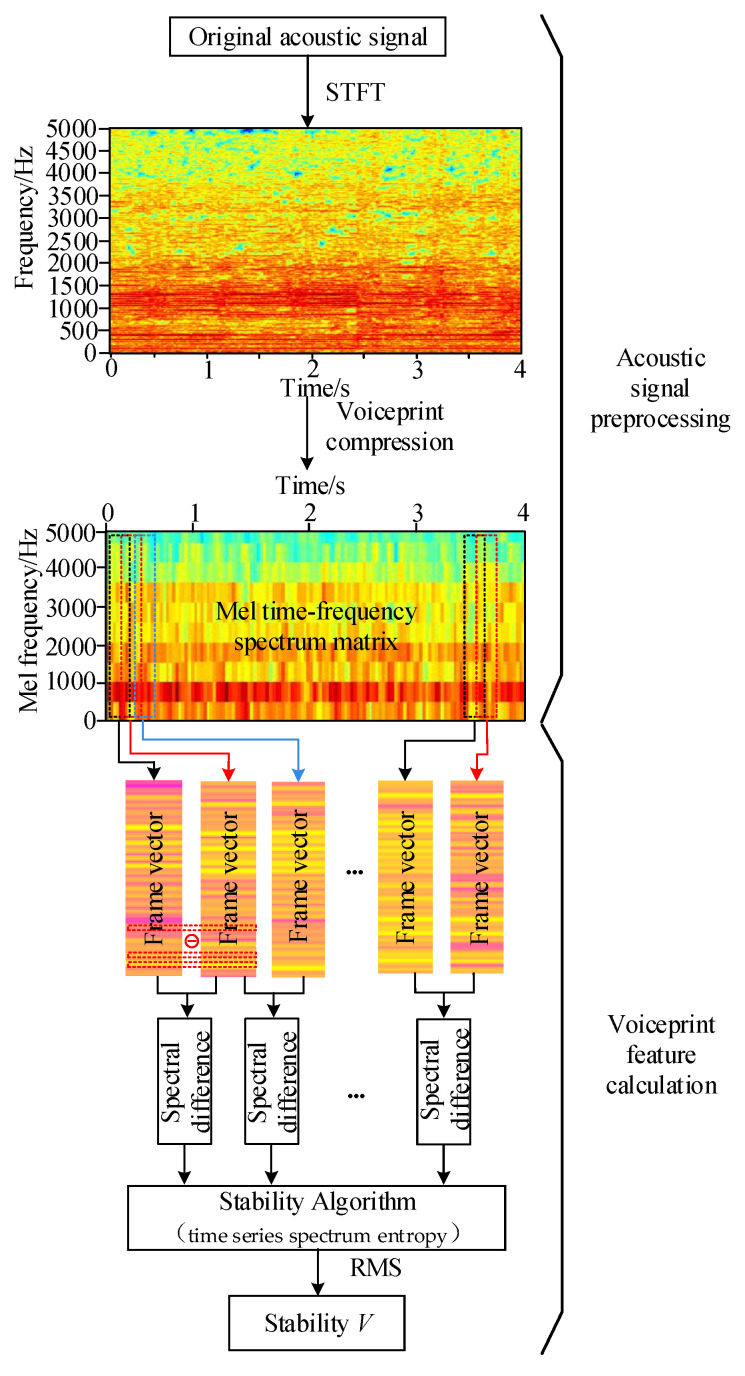
Calculation method for the stability of time-series spectral entropy.

**Figure 5 sensors-23-03258-f005:**
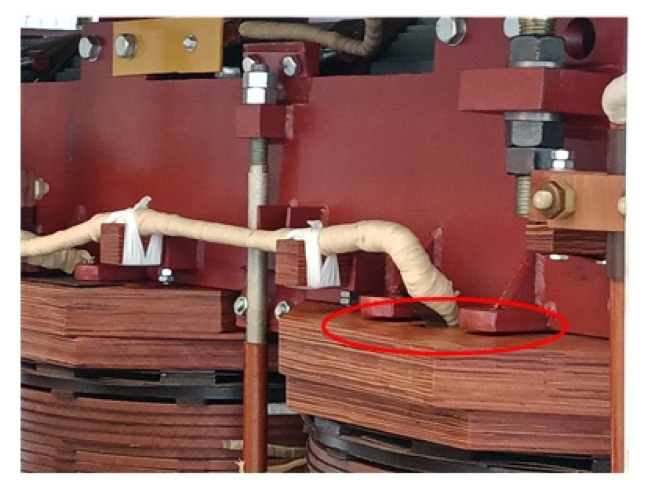
Diagram of transformer winding pad falling.

**Figure 6 sensors-23-03258-f006:**
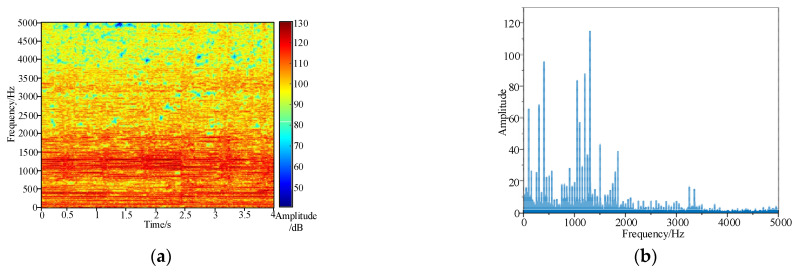
(**a**) Acoustic time–frequency spectrum; (**b**) spectrum distribution.

**Figure 7 sensors-23-03258-f007:**
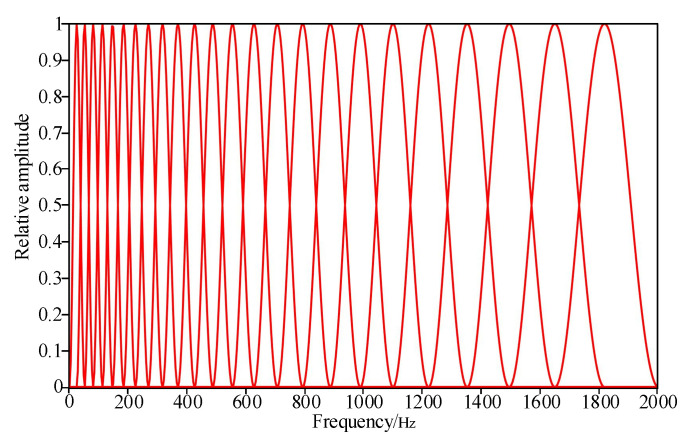
Transfer function of 24 Mel filters.

**Figure 8 sensors-23-03258-f008:**
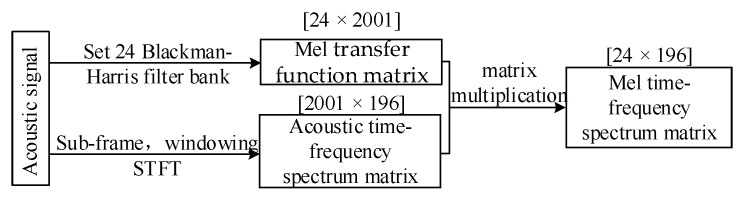
Calculation process of Mel spectrum.

**Figure 9 sensors-23-03258-f009:**
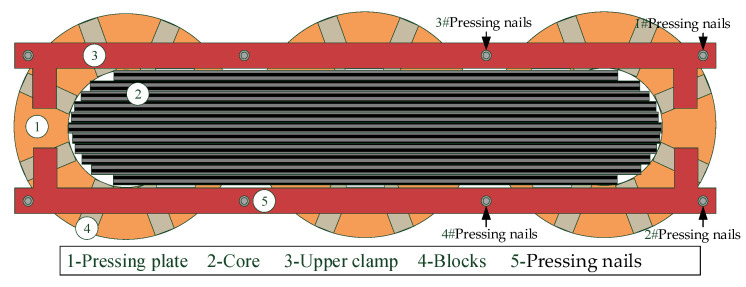
Loosening arrangement diagram of oil-immersed transformer winding.

**Figure 10 sensors-23-03258-f010:**
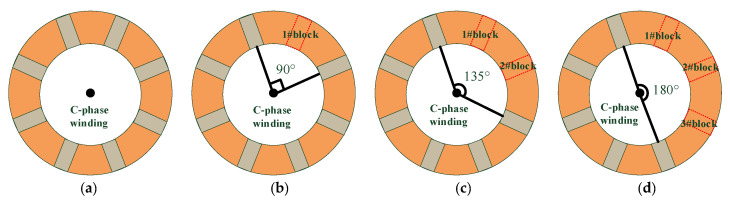
Schematic diagram of corresponding relationship between pad falling and loosening range. (**a**) Rated compression. (**b**) One piece falls off 90°. (**c**) Two pieces fall off 135°. (**d**) Three pieces fall off 180°.

**Figure 11 sensors-23-03258-f011:**
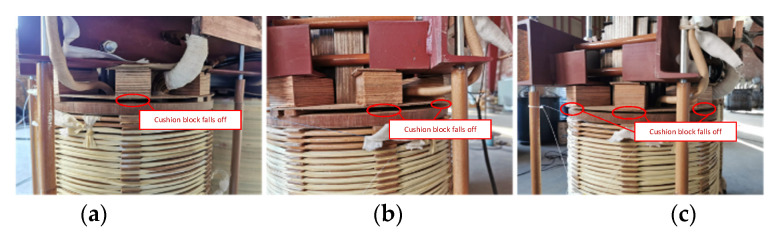
Site layout of pad falling in different loosening ranges. (**a**) 90°. (**b**) 135°. (**c**) 180°.

**Figure 12 sensors-23-03258-f012:**
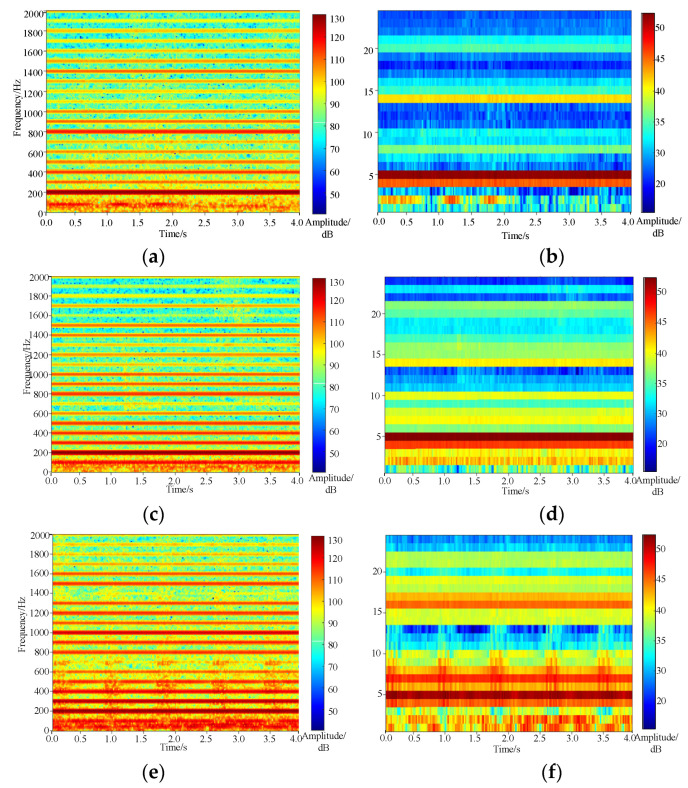
Time–frequency spectrum distribution of three operating states. (**a**) Acoustic time–frequency spectrum of rated compression (**b**) Mel time–frequency spectrum of rated compaction (**c**) Acoustic time–frequency spectrum when winding loosens (**d**) Mel time–frequency spectrum when winding loosens (**e**) Acoustic time–frequency spectrum when pad falls off (**f**) Mel time–frequency spectrum when pad falls off.

**Figure 13 sensors-23-03258-f013:**
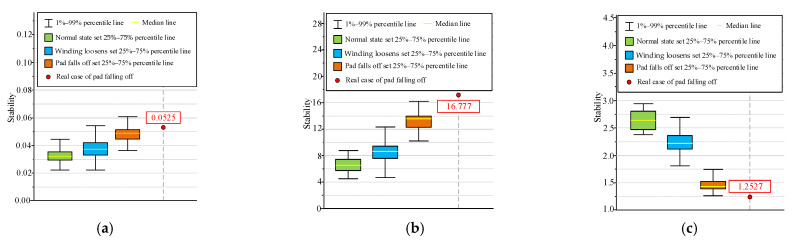
Comparison of various stability calculation methods under normal conditions, winding looseness and pad falling.(**a**) Euclidean distance. (**b**) Cosine distance. (**c**) Time-series spectral entropy.

**Figure 14 sensors-23-03258-f014:**
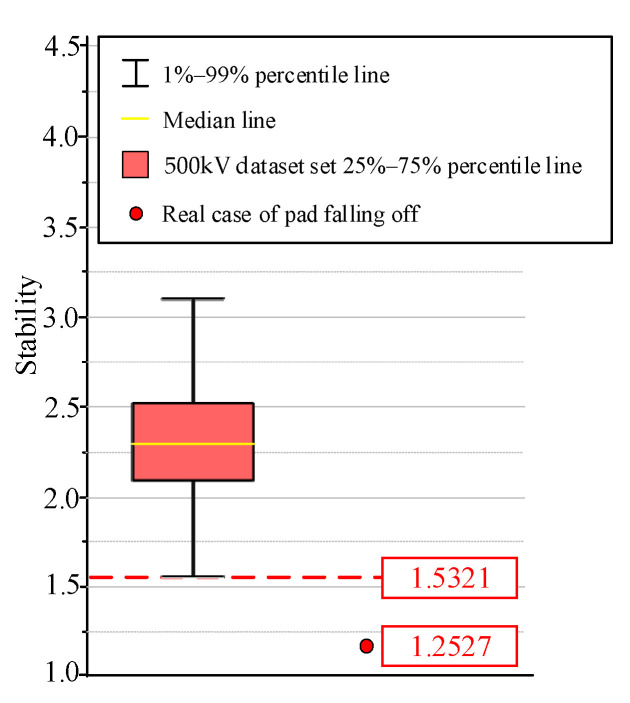
Spectrum distribution diagram of 500 kV dataset under operation status.

**Table 1 sensors-23-03258-t001:** Technical parameters of oil-immersed transformer.

Parameter	Value
Model	SY-400/10
Rated capacity	400 KVA
Rated voltage	10,000/400 V
Rated current	23.1/577 A
Connection group	Yy0
Rated frequency	50 Hz three-phase
Cooling mode	ONAN

**Table 2 sensors-23-03258-t002:** Looseness test scheme of winding.

Group	1# Press Nail Tightness	2# Press Nail Tightness	3# Press Nail Tightness	4# Press Nail Tightness
1	100%	100%	100%	100%
2	75%	100%	100%	100%
3	50%	100%	100%	100%
4	0%	100%	100%	100%
5	0%	75%	100%	100%
6	0%	50%	100%	100%
7	0%	0%	100%	100%
8	0%	0%	75%	100%
9	0%	0%	50%	100%
10	0%	0%	0%	100%
11	0%	0%	0%	75%
12	0%	0%	0%	50%
13	0%	0%	0%	0%

**Table 3 sensors-23-03258-t003:** Test scheme for winding end pad falling and combined winding loosening.

Group	Looseness Range of Winding	1# Press Nail	2# Press Nail	3# Press Nail	4# Press Nail
1	0°	100%	100%	100%	100%
2	90°	100%	100%	100%	100%
3	90°	0%	100%	100%	100%
4	90°	0%	100%	0%	100%
5	135°	0%	100%	0%	100%
6	135°	0%	0%	0%	100%
7	180°	0%	0%	0%	100%
8	180°	0%	0%	0%	0%

## Data Availability

The data presented in this study are available upon request from the first author. The data are not publicly available due to intellectual property protection.

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
