# Peer review of "Defect Identification Method for Transformer End Pad Falling Based on Acoustic Stability Feature Analysis"

_sensors, 2023, doi:10.3390/s23063258_

Round 1

Reviewer 1 Report

The language exepression should be improved in the paper.

Author Response

Thank you for your feedback. We have made adjustments and optimizations to the English descriptions in the manuscript.

Reviewer 2 Report

The paper is current, and published results are interesting.

The benefit of the paper is analyzes the vibration process of the fault from the perspective of mechanism and acoustic of transformer pad falling fault. The some issues must be addressed before publication.

The comments are as follow:

1. The References section used little sources.

2. The Introduction section must be rewritten. In this section authors must cite more related sources.

3. The Conclusions section should be improved with the most important obtained results (its exact values). Also the Conclusions seem to be too descriptive and general, without any details obtained in the presented analysis.

4. Please rewrite the explanation in the figures 2, 13, 14 – the English language not identifies some symbols.

Reviewer 3 Report

The authors address the following topic: Method of identification of transformer end plate drop defects based on the analysis of acoustic stability characteristics.

The work is novel and meets the research requirements.

Consider the correction of the labels of the figures, put it in English. Mainly 2, 13 and 14.

Put the results in a better way, in such a way that it is easy for the reader to follow the information.
